# Molecular basis of TASL recruitment by the peptide/histidine transporter 1, PHT1

Tânia F. Custódio[1,2], Maxime Killer[1,2,3], Dingquan Yu [1,2,3], Virginia Puente[1,2], Daniel P. Teufel[4], Alexander Pautsch[4], Gisela Schnapp [4], Marc Grundl[4], Jan Kosinski [1,2,5] & Christian Löw [1,2,6] ✉

PHT1 is a histidine /oligopeptide transporter with an essential role in Toll-like receptor innate immune responses. It can act as a receptor by recruiting the adaptor protein TASL which leads to type I interferon production via IRF5. Persistent stimulation of this signalling pathway is known to be involved in the pathogenesis of systemic lupus erythematosus (SLE). Understanding how PHT1 recruits TASL at the molecular level, is therefore clinically important for the development of therapeutics against SLE and other autoimmune diseases. Here we present the Cryo-EM structure of PHT1 stabilized in the outward-open conformation. By combining biochemical and structural modeling techniques we propose a model of the PHT1-TASL complex, in which the first 16 N-terminal TASL residues fold into a helical structure that bind in the central cavity of the inward-open conformation of PHT1. This work provides critical insights into the molecular basis of PHT1/TASL mediated type I interferon production.

Proton coupled oligopeptide transporters (POTs) are responsible for the uptake and distribution of nutrients such as di- and tripeptides into the body and regulating metabolism[1–3]. They also play key roles in human diseases, impacting the pharmacokinetic profiles of orally administered drug molecules[4]. They belong to the major facilitator superfamily (MFS) and share a common fold[5–7], which is comprised of twelve transmembrane helices with intracellular amino- and carboxy-termini and a pseudosymmetry between the structural motifs. In humans, the POT family contains five members with distinct expression profiles, substrates and biological function (Supplementary Fig. 1). Peptide transporter 1 (PepT1 = SLC15A1) and Peptide transporter 2 (PepT2 = SLC15A2) are the best studied members in the POT family, responsible for the uptake of nutrients mainly in the small intestine and kidney, respectively[8–10]. Both are plasma membrane transporters and share around 70 percent sequence similarity. They are known for their substrate promiscuity, as they can transport a large variety of di- and tri-peptides but also several clinically relevant drugs[11–13]. PHT1 (SLC15A4) and PHT2 (SLC15A3) are found intracellularly, in the endo-lysosome membrane of immune cells and they were characterized by

their capacity to transport histidine, short peptides and bacterially-derived peptides such as the muramyl dipeptide (MDP) and other nucleotide-binding oligomerization domain-containing protein 1 and 2 (NOD1, NOD2) ligands[14–20]. SLC15A5 is predominantly expressed in the brain and has not been functionally characterized yet[21]. Despite appearing to be functionally indistinguishable, PHT1 and PHT2 have distinct cellular expression patterns. PHT1 is mostly expressed in the plasmacytoid dendritic cells (pDCs) and B cells[20,22,23], whereas PHT2 is predominantly expressed in macrophages[15].

In addition to histidine and peptide translocation, PHT1 has been found to play an important role in Toll-like receptor-mediated production of type I interferon (IFN-I)[23–26]. Endolysosomal Toll-like receptors (TLRs) are crucial players in the innate immune system by sensing nucleic acids derived from foreign pathogens and recruit specific adaptor proteins to promote the expression of pro-inflammatory cytokines, chemokines and IFN. Persistent stimulation of these inflammatory genes is the central cause of several auto-immune diseases. In fact, different studies have associated PHT1 with systemic lupus erythematosus (SLE)[3,5–7], inflammatory bowel diseases[27]

[1]Centre for Structural Systems Biology (CSSB), Notkestraße 85, 22607 Hamburg, Germany. [2]European Molecular Biology Laboratory (EMBL) Hamburg, Notkestraße 85, 22607 Hamburg, Germany. [3]Collaboration for joint PhD degree between EMBL, and Heidelberg University, Faculty of Biosciences, 69120 Heidelberg, Germany. [4]Boehringer Ingelheim Pharma, Birkendorferstraße 65, 88397 Biberach, Germany. [5]Structural and Computational Biology Unit, European Molecular Biology Laboratory, Meyerhofstraße 1, 69117 Heidelberg, Germany. [6]Twitter: @AllUNeedIsLoew ✉e-mail: christian.loew@embl-hamburg.de

and type 2 diabetes[28]. Because PHT1 is a proton-coupled histidine transporter, absence of function dysregulates the lysosomal conditions, including histidine concentrations and acidity, thereby impeding the function of endolysosomal TLRs (TLR3, 7–9). Furthermore, PHT1 transport activity was shown to be required for the TLR7-IFN-I induced activation of the mTOR pathway[26].

Besides the regulation of the lysosomal environment, a recent study by Heinz et al.[29] was able to position PHT1 downstream the TLR pathway, via a newly-identified adaptor protein. This adaptor protein encoded by the CXorf21 gene was dubbed 'TLR adaptor interacting with SLC15A4 on the lysosome' (TASL). Only in complex with PHT1, TASL can recruit and activate interferon regulatory factor 5 (IRF5), which in turn stimulates the production of IFN-I genes (Supplementary Fig. 2a). TASL is predicted to be mainly intrinsically disordered with only short helical structural elements at the C-terminus, where the conserved pLxIS motif (common to other IRF adaptor proteins) is found. Upon phosphorylation of the pLxIS motif, TASL is expected to bind and stimulate IRF5 phosphorylation-induced dimerization and consequently activation, in analogy with other adaptor proteins[30–32]. The N-terminal residues of TASL were shown to be critical for binding to PHT1[29]. However, the precise nature of the interaction between TASL and PHT1 is still largely unknown.

The peculiar position of PHT1 in TLR pathways is becoming increasingly attractive for the development of therapeutics for inflammatory disorders. On the one hand, the fundamental role of PHT1 in optimizing the lysosomal environment via its transport activity encourages the development of potent PHT1 inhibitors. On the other hand, its pivotal role in the IRF5 pathway and subsequent predisposition to SLE, urges the design of compounds that can mechanistically prevent the recruitment of the adaptor protein TASL by PHT1 and consequently IFN-I production. Structural information as well as biophysical analysis of PHT1 and its interaction with TASL in vitro will aid to the development of such therapeutics. The structures of the mammalian POT family members PepT1 and PepT2 have recently been determined and provided insights into the architecture and dynamics of this transporter family throughout the transport cycle[33–35]. However, extrapolating these findings to PHT1 is challenging due to PHT1 specific interaction with TASL and the low sequence identity (less than 20%) to their plasma membrane counterparts.

In this work, we determine the structure of chicken PHT1 together with a synthetic nanobody (sybody, Sb) in the outward-open conformation at 3.3 Å resolution by single particle cryo-EM. By combining in vitro, in vivo and in silico approaches, we map and characterize the PHT1-TASL interface. Our data illustrate that TASL binds into the central cavity of the inward-open conformation of PHT1, with both N- and C-domains clamped around the first 16 N-terminal TASL residues, that fold into a helical structure upon complex formation. PHT1 binds TASL in vitro with an affinity of approximately 360 nM and we identify several residues that are critical for this interaction. Our PHT1-TASL model as well as the PHT1 outward-open structure can serve as a guide in the development of molecules to either block the transporter or inhibit TASL binding to prevent IRF5 downstream signaling.

## Results

### PHT1 is a bona fide peptide transporter

PHT1 has been extensively characterized for its capacity in transporting histidine, di- and tripeptides, bacterial peptidoglycan components and several histidine precursors, such as carnosine[16,18,19,27,36]. Different cell-based assays were used to monitor uptake, which could be the reason for several inconsistencies found among published work[16,18,36,37]. For example, the first study on substrate selectivity of PHT1 was performed in Xenopus Laevis oocytes and the uptake of histidine ($K_m = 17\,\mu M$) and carnosine with maximum activity at pH 5.5 could be detected. However, the subcellular location of PHT1 was not considered and its expression at the plasma membrane was not

confirmed[37]. Another study in 2005 by Bhardwaj et al.[18], used the transient transfection method for the expression of PHT1 in COS-7 cells, and the authors described the uptake of histidine and carnosine but not of glycylsarcosine (GlySar), a known PepT1 substrate. Again, the authors did not show the expression of PHT1 at the plasma membrane. In a more recent study[16], a plasma membrane mutant of PHT1 was stably transfected in MDCK cells and the uptake of Histidine, carnosine and also GlySar was reported. The substrate uptake was pH dependent with a maximum activity at pH 6.5. Furthermore, histidine inhibition studies from the same study yielded an $IC_{50}$ value for histidine greater than 1 mM, considerably different from the reported $K_m$ value. The observed variations between the published studies might be explained by different expression systems or different type of methodology used (i.e., transient vs stable transfections) and arguably the presence of other endogenous transport systems.

To examine the substrate specificity of PHT1 in vitro, we transiently expressed chicken PHT1 (which will be referred to as PHT1 from here on), sharing 84 % sequence similarity and 77% sequence identity with human PHT1 (Supplementary Fig. 1), in human embryonic kidney 293-F (HEK293-F) cells, and purified the protein in detergent solution. We tested binding of PHT1 to a variety of different amino acids, and di- and tripeptides at a concentration of 5 mM, by the thermal shift method using differential scanning fluorimetry (DSF)[38–40]. Upon binding of a potential substrate, the transporter is stabilized against heat induced unfolding resulting in an increased melting temperature (Fig. 1a). Based on these data, we show that PHT1 exhibits a rather broad substrate specificity typical for POTs and the transporter is capable of binding both, di- and tripeptides. From the two bacterial-derived peptides, only N-Formylmethionyl-leucyl-phenylalanine (fMLP) but not the NOD2 ligand, muramyl dipeptide (MDP), could bind PHT1. However, neither L-histidine nor any of the histidine containing di- and tripeptides, including the histidine precursors (carnosine and anserine), were capable of stabilizing PHT1 (Fig. 1a). Histidine binding was also tested in a pH dependent manner, but no stabilization effects could be observed (Supplementary Fig. 2b). Based on the analysis of a concentration-dependent thermal shift experiment, we determined the binding affinities of PHT1 for two sets of dipeptides, Lysine-Valine (KV) and Arginine-Phenylalanine (RF), at $63 \pm 11\,\mu M$ and $73 \pm 12\,\mu M$, respectively (Supplementary Fig. 2c, d). Notably, recombinantly expressed PHT1 is heavily glycosylated (Fig. 1b) but thermal stability of the protein is not affected by the glycosylation composition (Supplementary Fig. 2e).

To better understand the substrate specificity/preference within the POT family, we have screened 65 di- and tripeptides against PHT1, PepT1 and PepT2 at two different pH values (Supplementary Fig. 3). The plasma membrane transporters are stabilized by a broader set of peptides than PHT1. PepT2 as a known high-affinity and low-capacity transporter shows higher stabilization effects upon peptide binding, compared to PepT1 or PHT1. For all three tested transporters, peptides that stabilize the protein at pH 7.5, also stabilize it at pH 5.5, with the exception of a few dipeptides containing positive residues, like Lysine-Alanine (KA), Lysine-Proline (KP), Lysine-Lysine (KK) and Arginine-Proline (RP). This indicates that a protonation event in one or more residues that are part of the binding site of these transporters negatively influences binding of these positively charged peptides. Overall, peptides with negatively charged residues do not strongly stabilize any of the transporters. In conclusion, the substrate preference of PHT1 seems to be more restricted compared to PepT1 or PepT2.

### Single particle cryo-EM structure of PHT1 in complex with Sb27

Following previous strategies that allowed us to determine the structures of human PepT1 and PepT2[33], we first attempted to image PHT1 purified in the detergent DDM-CHS. However, PHT1 lacks distinct structural features outside the transporter unit (contrary to PepT1 and PepT2) and projections failed to be precisely aligned. To overcome

these limitations, we selected synthetic nanobodies (sybodies) as fiducial markers and with the potential to stabilize a particular conformation[41]. Overexpression of the plasma membrane mutant of PHT1 (PHT1 L12A/L13A), yielded higher amounts of protein compared to the construct expressing at the lysosome, and was selected for structural studies. Sybody selections on deglycosylated PHT1 L12A/L13A (PHT1 overexpressed in Expi293FGnT1- cell line and treated with EndoH) were carried out with three sybody libraries (Supplementary Fig. 4), following established procedures[42,43]. Among the identified hits (Supplementary Table 1), one binder termed sybody 27 (Sb27) formed a stable complex (Supplementary Fig. 4c) and stabilized PHT1 by six degrees (Fig. 1c). We used a biolayer interferometry (BLI) assay to kinetically characterize the interaction between Sb27 and PHT1. Sb27 exhibited a fast on- and off-rate resulting in a moderate affinity of approximately 400 nM (Fig. 1d).

Despite the low affinity of Sb27 towards PHT1, cryo-EM data on the PHT1-Sb27 complex immediately resulted in improved 2D class averages with clear secondary structure features, compared to the transporter alone (Supplementary Fig. 5a, b). Surprisingly, two distinct characteristics at the periphery of the detergent micelles were visible from the 2D classes. We obtained a 3D reconstruction for PHT1 bound to two copies of Sb27 on the cytosolic side at a nominal resolution of 3.3 Å (Fig. 2, Supplementary Figs. 5–7 and Supplementary Table 2). The EM map allowed model building of the twelve transmembrane helices, statring from residue 19 at the N-terminus until 569 at the C-terminus (out of 581 residues). A stretch of the

bundle bridge (residue 279–300) and two loops at the lumen side (residues 135–159 and 362–370) connecting TM3/TM4 and TM7/TM8 were poorly resolved and could not be modeled (Fig. 2 and Supplementary Fig. 6). The overall structure of PHT1 adopts the classical MFS fold, where the N-terminal (N-domain, TM1–6) and C-terminal domain (C-domain, TM 7–12) enclose a large cavity that opens to the lumen side, a state defined as outward-open (Fig. 2c). These domains are connected by the bundle bridge comprised of two intracellular amphipathic helices. On the luminal side two short β-strands link TM9 with TM10, which in homologous PepT1 and PepT2 corresponds to the position of the extracellular domain (ECD) (Supplementary Figs. 1 and 8a). Access towards the cytosol is closed, caused by several tight interactions between the N- and C-domain. In particular, a salt bridge is formed between Arg200 and Glu482. Additional polar interactions further stabilize this conformation, such as Arg200 to the main chain carbonyl of Pro407, Asp189 to Tyr485, Lys98 to Gln493 and from Asp93 to the side chain and main chain nitrogen of Ser494 (Fig. 2d).

A structural overlay of PHT1 with the outwardopen state structure of human PepT1[33] (PDB ID: 7PMX) reveals that the transporter parts of both structures are very similar (RMSD Cα is 1.3 Å over 310 residues) (Supplementary Fig. 8b). Some local rearrangements of the N-domain can be discerned while the C-domain overlays entirely. A comparison of the substrate binding pocket depicts a similar arrangement of highly conserved residues in PHT1 and PepT1 (Supplementary Figs. 1 and 8b). With the exception of Asp381 (corresponding to an asparagine residue

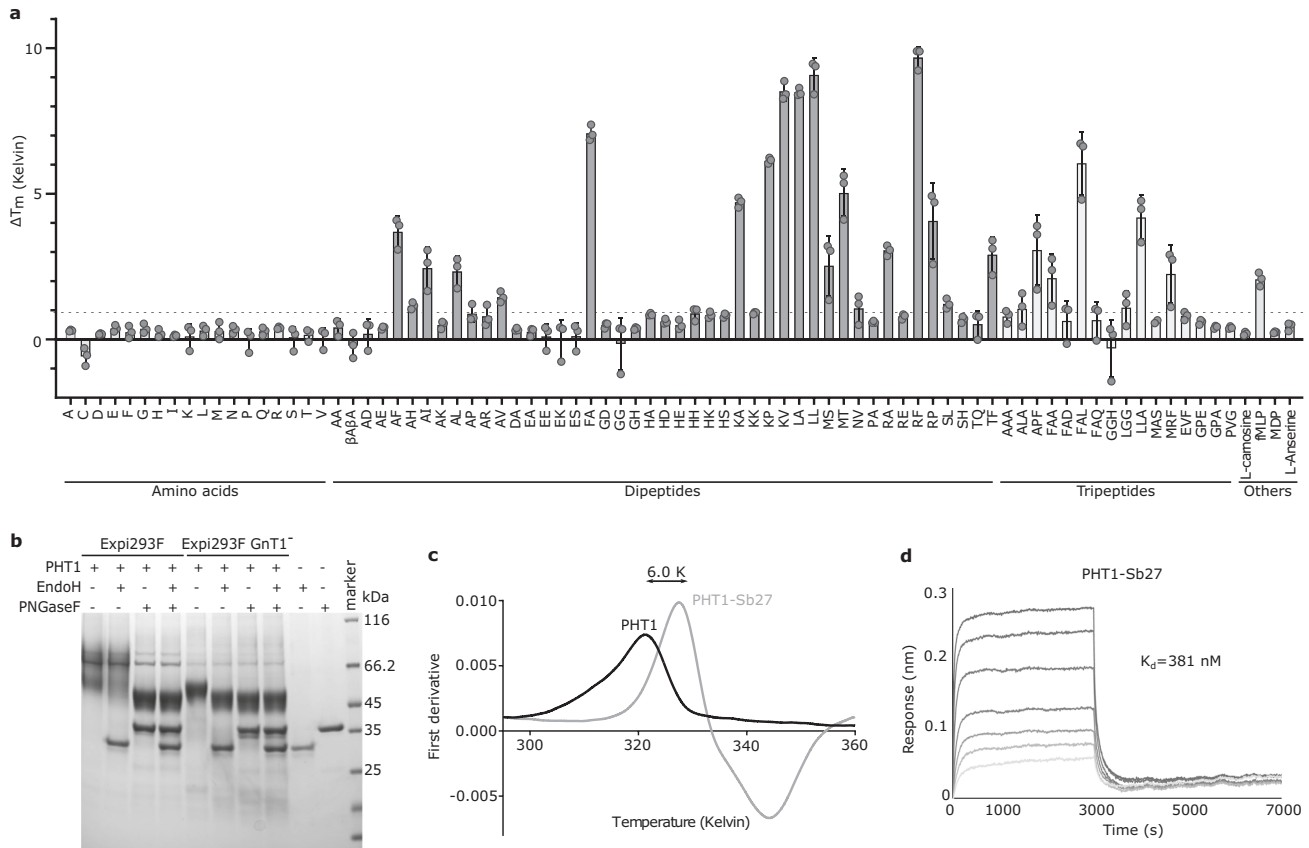

**Fig. 1 | Functional characterization of PHT1. a** Thermal stability of PHT1 screened with a ligand library at a concentration of 5 mM. Changes in the melting temperature of PHT1 in the presence of a ligand compared to the melting temperature of PHT1 alone (ΔT_m) are shown. Library is noted using the 1 letter abbreviation for amino acids. Data represent the mean ± SD of three biological replicates. Binding of histidine or histidine containing peptides was not detected. **b** SDS-PAGE gel of PHT1 expressed in different cell lines and deglycosylation treatment by PNGaseF or EndoH enzymes. One representative measurement is shown. Three independent measurements were performed and showed similar results. **c** First derivative of the F_350:330 signal of PHT1 thermal unfolding data and in complex with Sb27. One representative measurement is shown. Two independent measurements were performed and showed similar results. **d** BLI sensorgrams of immobilized PHT1 with 2-fold serial dilution of 800 nM Sb27. Source data for relevant information are provided as a Source Data file.

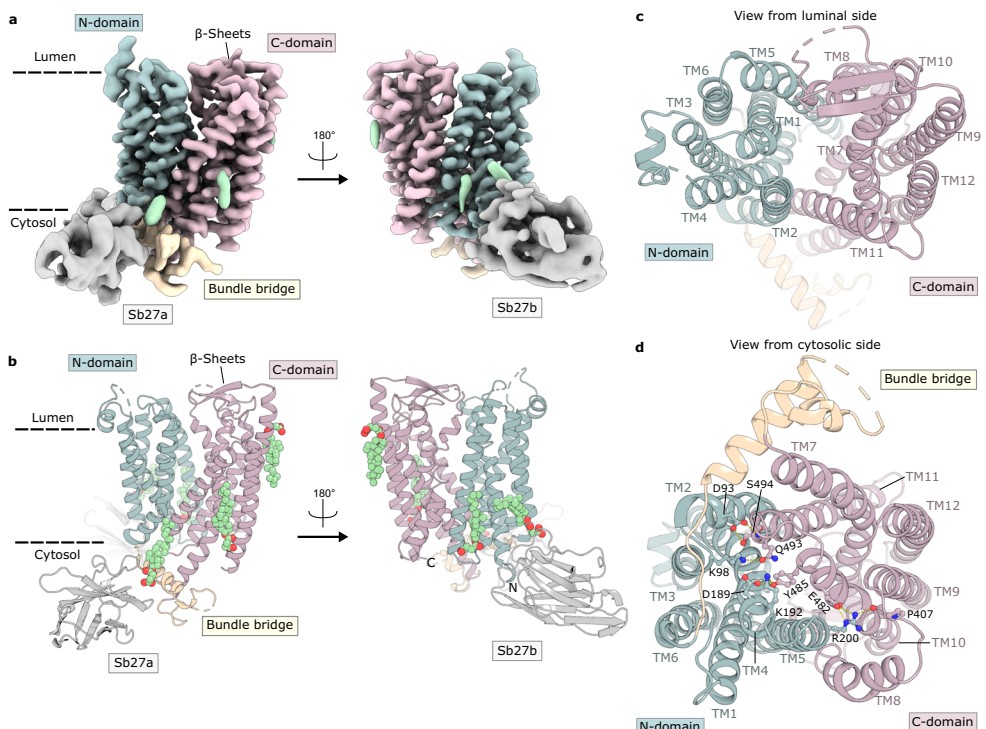

**Fig. 2 | Outward-open structure of PHT1 in complex with Sb27. a** Cryo-EM map and **b** cartoon representation of PHT1 in complex with Sb27. The structure represents the outward-open conformation of PHT1 in complex with two molecules of Sb27 (gray). Sb27a binds at the bundle bridge of PHT1 (yellow) which links the N-domain (blue) with the C-domain (pink) of PHT1, while Sb27b binds at the N-domain of PHT1. **c** View of PHT1 from the luminal side, showing a cavity open to the lysosome lumen, while **d** view from the cytosol side illustrates that the cavity is closed and held in place by several highlighted interactions between the N- and C-domains.

in human PepT1 and human PepT2), all critical residues involved in the coordination of the N- and C- termini of di- and tripeptides are conserved, explaining the promiscuity and broad substrate specificity within this family[33–35].

By visual inspection of the EM map, we identified five additional elongated densities close to the transmembrane domains of PHT1 (Fig. 2 and Supplementary Fig. 6). Since PHT1 was purified in the presence of cholesteryl hemisuccinate (CHS), we speculate that the observed densities may originate from bound CHS molecules. Lipids and cholesterol or cholesterol mimics, such as CHS, are known to be important in maintaining the structure and function of membrane proteins and are often found in experimental structures of membrane proteins[44–46]. Indeed, CHS molecules fit the densities well and provide an explanation for the two distinct epitopes on PHT1 (Supplementary Fig. 7). In both epitopes, the interaction is mediated by a CHS molecule which is sandwiched between the transporter and Trp103 from the CDR3 loop of Sb27.

### Interaction between PHT1 and the adaptor protein TASL

To decipher the molecular interactions between PHT1 and the adaptor protein TASL, we first conducted pull-down assays of PHT1 co-expressed with chicken TASL (which will be referred to as TASL from here on) in Expi293F cells to confirm their direct interaction. In agreement with previous data[29], TASL was efficiently pulled down by endosomal PHT1 or a PHT1 mutant (PHT1 L12A/L13A) expressed at the plasma membrane, but not human PepT2 (Fig. 3a). Our attempts to purify larger quantities of the full-length PHT1-TASL complex for structural studies failed, mainly due to low expression levels and severe proteolytic degradation of TASL. Given the difficulties in purifying the intact complex and the recent breakthrough in the accuracy of predicting protein structures by AlphaFold[47], we used AlphaFold-multimer[48] through the AlphaPulldown pipeline[49] to predict the

structure of the PHT1-TASL complex (Fig. 3b, Supplementary Fig. 9 and Supplementary Table 3).

As outlined above, TASL is an intrinsically disordered protein (IDP) with a three-helix bundle arrangement at the C-terminus, before the conserved pLxIS motif (Supplementary Fig. 9a). The N-terminus, shown to be important for binding to PHT1, appears to be disordered. This is different for the predicted PHT1-TASL complex (Fig. 3b and Supplementary Fig. 9b and c). In line with all obtained models, the N-terminal residues of TASL fold into a helical structure (up to residue 34) when bound to PHT1 (Supplementary Fig. 9d). The N-terminal helix of TASL binds in the central cavity of the inward-open conformation of PHT1, between the N- and C-domains (Fig. 3b, c). The tight binding interface comprises the first 16 residues of TASL and the overlay of the ten best scoring AF2 models illustrates a defined position of these residues inside the transporter cavity. Beyond this, the C-terminal part of this α-helix displays a certain degree of flexibility or model uncertainty (Supplementary Fig. 9d). A comparison of different species of TASL sequences identified the first ten residues as conserved (Fig. 3d) and based on the PHT1-TASL AF2 model, we designed a peptide comprising the first 13 residues of TASL (TASL$_{1-13}$) to probe the interaction by biophysical tools. Binding of the TASL$_{1-13}$ peptide induced a strong stabilization effect on the melting temperature of PHT1. Concentration-dependent thermal shift assays allowed us to determine the binding affinity to be 360 nM (Fig. 3e). These experiments are in good agreement with microscale thermophoresis (MST) binding data using a fluorescently labeled TASL$_{1-13}$ peptide (Supplementary Fig. 10a). TASL interaction is conformation dependent, and only possible with the transporter in the inward-open state with the substrate binding cavity open towards the cytosol and closed to the lysosomal lumen (Fig. 3c). Sb27 stabilized PHT1 in the outward-open conformation by forming an interaction network with the N-domain of PHT1. Cryo-EM reconstructions of a potential PHT1-Sb27-TASL$_{1-13}$ complex

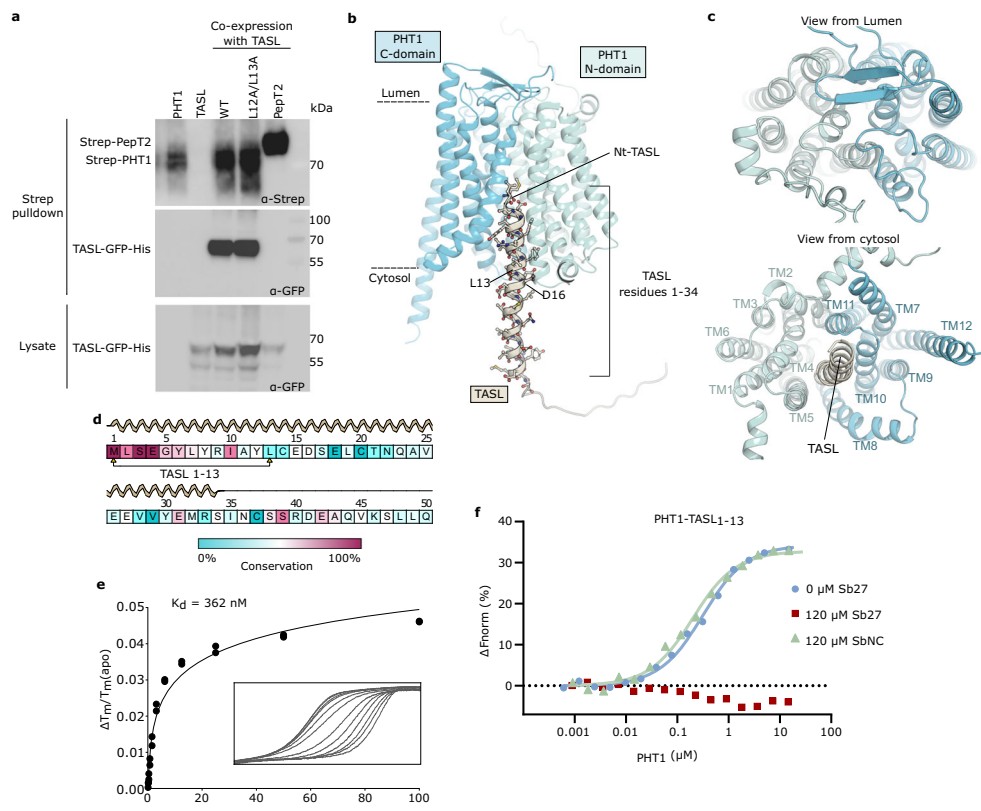

**Fig. 3 | Interaction between PHT1 and TASL. a** Binding analysis of PHT1 and TASL. TASL was pulled down by Twin-Streptavidin-tagged PHT1 using strep-tactin beads, and the eluates were analyzed by western blot. Data are representative of three independent experiments with similar results. **b** Cartoon representation of the AF2 model of PHT1 (blue) in complex with TASL (brown). TASL N-terminal residues form a helical structure that binds in the central cavity of the inward-open conformation of PHT1, with a cavity closed to the luminal side (**c**). **d** Sequence conservation of the first 50 residues of TASL, derived from 289 homologs found across all mammal species. **e** Thermal unfolding titration of TASL$_{1-13}$ with PHT1. Data

extrapolated from the F$_{350}$ signal were fitted according to a modified script of Hall's method and normalized unfolding transitions with increasing concentrations of the peptide are shown. Data represent two replicate experiments. **f** Microscale thermophoresis (MST) binding data of PHT1 with fluorescently labeled TASL$_{1-13}$ peptide, in the presence or absence of 120 μM Sb27 or a negative control sybody. One representative measurement is shown. Three independent measurements were performed and showed similar results. Source data for relevant information are provided as a Source Data file.

resulted in the PHT1-Sb27 complex only, since access to the binding site for TASL was blocked. This has been further confirmed by monitoring binding of the fluorescently labeled TASL peptide to PHT1 in the presence of a constant concentration of Sb27 or an unrelated sybody, used as a negative control (SbNC). Our data show that the TASL peptide can only bind to PHT1 in the presence of SbNC but not in the presence of Sb27 (Fig. 3f).

**Molecular basis of TASL binding to PHT1**

A more detailed analysis of the PHT1-TASL binding interface uncovered that the first four TASL residues are coordinated similarly to a natural substrate by PHT1 (Fig. 4a). The N-terminal amino group of TASL is in salt-bridge-forming distance to the strictly conserved Glu473 residue and Tyr343 of PHT1. Arg44 and Lys180 of PHT1 are predicted to constitute a salt bridge with the side chain of TASL Glu4, similarly as they would do with the carboxy-terminus of a di- and tripeptide. Glu473 of PHT1 makes additionally polar contacts to the main chain amino group and side chains of Ser3 of TASL. The methionine sulfur-aromatic interaction between TASL Met1 and PHT1 Tyr82 further stabilizes the tight binding (Fig. 4a, Supplementary Figs. 1, 8 and 10). The other TASL interacting residues (residues 5–16) are held in place by the N- and C-domains of PHT1 largely through hydrophobic contacts, in particular via Met497, Phe500, Phe501 and Y485 at the C-domain (Fig. 4a and Supplementary Fig. 10c, d). All residues identified as part of the interface in PHT1 are highly conserved (Supplementary Fig. 11).

To confirm and validate the interface identified in the AF2 model analysis, we designed TASL peptides with different properties and measured the stability effect on the PHT1 melting temperature (Fig. 4b, Supplementary Fig. 12a and Supplementary Table 4). First, we investigated the binding of peptides of different lengths. PHT1 stabilization was only detectable for a TASL peptide constituting the first 11 residues and becomes more pronounced with the addition of further residues. Tightest interactions were observed for TASL peptides with lengths of 13 and 14 residues. Concentration dependent titrations yielded a similar affinity (approximately 400 nM) for these peptides. Increasing the peptide length to 15, 17 or 21 residues did not further improve the affinity towards PHT1. Extending the TASL$_{1-13}$ peptide at the N-terminus by a single glycine or two serine residues decreased or abolished binding, while the addition of multiple glycine and serine residues at the C-terminus of the peptide did not affect the interaction. Deletion of the first residues or substitutions by alanine residue decrease affinities almost 25-fold or 35-fold, respectively (Supplementary Fig. 12a and Supplementary Table 4). Mutation of the highly conserved TASL Glu4 leads to a complete disruption of the interaction, highlighting the importance of Glu4 on TASL binding to PHT1 (Fig. 4b). In addition, competition assays confirmed that the TASL peptide and RF or KV dipeptides have overlapping binding sites with PHT1 in agreement with the PHT1-TASL model (Supplementary Fig. 12b).

To correlate the obtained in vitro binding data of the TASL peptides with full-length TASL, we conducted pull-down assays

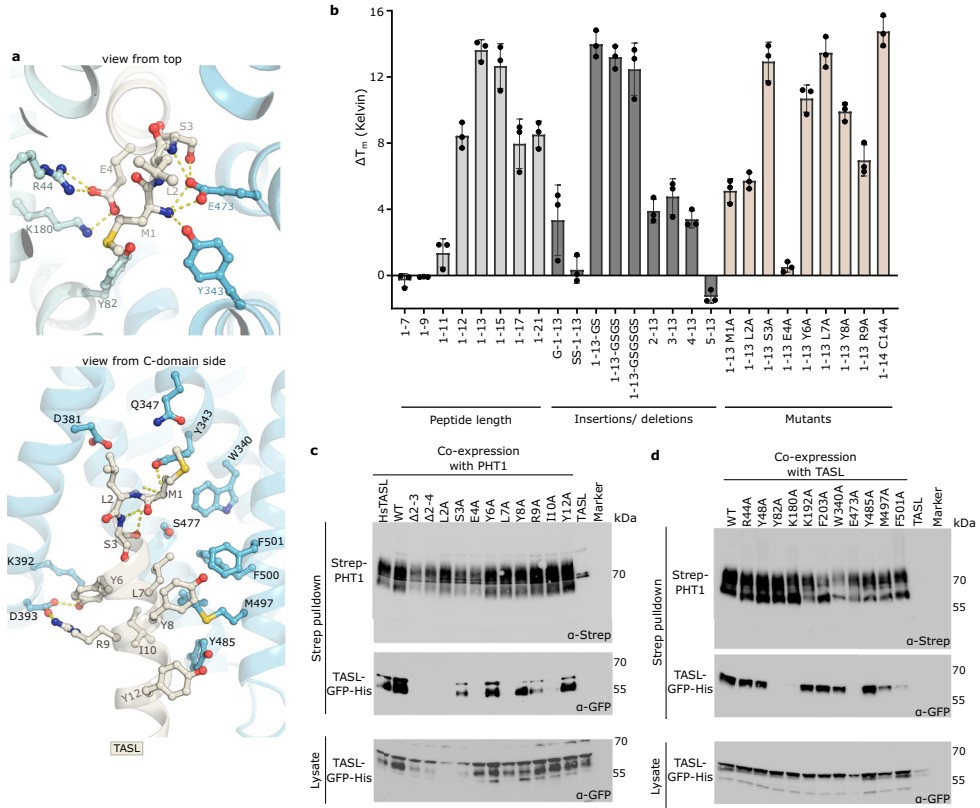

**Fig. 4 | Molecular mechanism of TASL recognition by PHT1. a** Close-up view of the central binding pocket (top) and C-domain interface of PHT1 (bottom), in the AF2 model of PHT1-TASL. Selected residues are shown as sticks and yellow dashes indicate hydrogen bonds (cutoff at 3.2 Å). **b** Thermal stability of PHT1 screened with different TASL peptides at a concentration of 50 μM. Changes in the melting temperature of PHT1 in the presence of a peptide as compared to the melting temperature of PHT1 alone ($\Delta T_m$) are shown. Data represent the mean ± SD of three biological replicates. **c** Pull-down assay using Twin-Streptavidin-tagged PHT1 and GFP-His-tagged TASL mutants at the interface residues. Shown data are representative of two independent experiments with similar results. **d** Pull-down assay using Twin-Streptavidin-tagged PHT1 mutants and GFP-His-tagged TASL at the interface residues. Shown data are representative of two independent experiments with similar results. Source data for relevant information are provided as a Source Data file.

using Twin-Streptavidin-tagged PHT1 and TASL mutants (Fig. 4c and Supplementary Fig. 13a). Most mutations of the first 10 residues or deletions in TASL abolished or decreased the amount of TASL protein pulled down by PHT1. The experimental data are in line with the in vitro binding data and support the model that the first 13 TASL residues are sufficient for binding PHT1 without other crucial interaction sites. Furthermore, we also showed that full-length human TASL (HsTASL) is efficiently pulled-down by chicken PHT1, highlighting once again the role of the conserved, first N-terminal residues, of TASL. This also agrees with pull-down data, where a GFP construct fused N-terminally with the first 30 residues of TASL tightly interacts with PHT1 (Supplementary Fig. 13a). Mutations upstream of Tyr12 had little or no effect on binding similarly as observed in the thermal shift data (Fig. 4c and Supplementary Fig. 13a). To challenge the TASL interaction from the PHT1 side, we performed the same pull-down assay with wild type TASL, but mutated eleven residues in PHT1, which could play a role in the coordination of TASL. We identified five single point mutations (Lys180, Glu473, Tyr82, Met497, Phe501) in PHT1 which led to a complete or severe loss of TASL binding consistent with the PHT1-TASL model. The residues in PHT1 that abolished TASL binding (Lys180, Tyr82 and Glu473) are part of the conserved substrate binding pocket in the POT family and coordinate the first four residues of TASL. Residues Arg44, Tyr48 and Trp340 are in close proximity to the first four residues of TASL, but mutating those in PHT1 to an alanine (or a glutamate residue in the case of Arg44), did not have a profound effect on TASL binding (Fig. 4d and Supplementary Fig. 13b), indicating a supportive role of these residues in the interface of the complex.

Met497 and Phe501 are part of the hydrophobic interaction surface at the C-domain of PHT1, which likely increase the affinity of PHT1 towards TASL, explaining the observed partial loss of binding. The contribution of the hydrophobic interaction of Phe203 at the N-domain of PHT1 is not as pronounced (just opposite to Met497 and Phe501 residues). Disruption of the salt bridge formed by Lys192 of PHT1 and Glu15 of TASL does not abolish binding of TASL, in line with the experimental data from the mutational studies of TASL. Lastly, we confirmed that TASL recruitment/binding is specific to PHT1 and not to other members of the POT family by both in vitro binding data of the TASL peptides and in vivo pull-down assays with full-length TASL (Supplementary Fig. 13c).

Together, these studies characterized and validate the PHT1-TASL interface identified in the AlphaFold2 model. While the 16 first residues of TASL are part of the complex interface, located in the inward-open cavity formed by the N- and C- domains of PHT1, the first 13 residues of TASL are sufficient for the high-affinity binding to PHT1. The coordination of the first four residues of TASL in the central binding pocket of PHT1 are imperative for the interaction, while the large hydrophobic surface from the C-domain of PHT1 boosts the affinity towards TASL.

## Discussion

The exact etiology of SLE is not well understood but is characterized by the excessive production of pathogenic autoantibodies and high levels of INF-I, which lead to tissue and organ damage[50]. B cells and pDCs play an important role in the pathogenesis of lupus, as aberrant stimulation of B cells through the endolysosome TLR pathway promotes loss of

tolerance[50]. Not surprising, the lysosomal proton coupled oligopeptide transporter PHT1, which is abundantly expressed in B cells and pDCs plays a pivotal role in the pathogenesis of SLE. While it has been demonstrated that PHT1 transport activity is important for its physiological role[17,23,26], it has also been discovered to be essential in the recruitment of the adaptor protein TASL and IRF5 activation[29]. Studies using feeble mice (which carry an SLC15A4 loss- of- function mutation) show amelioration of the disease[23,51]. Thus, PHT1 is a crucial target for therapeutic intervention in lupus and its functional and structural characterization is important to further elucidate its molecular position and in silico design of compounds that can block PHT1 activity and/or recruitment of TASL.

In this study, our data confirmed that PHT1 is a bona fide peptide transporter. The key residues of the peptide binding pocket are highly conserved except for Asp381 (corresponding to Asn residue in PepT1 and PepT2). We also determined the binding affinities between PHT1 and the two dipeptides, KV and RF, in the μM-range. These are in the same range as reported histidine transport affinities for human and rat PHT1[16,37] and the $IC_{50}$ value of the dipeptide Alanine-Phenylalanine (AF) for human PepT2[33].

Contrary to plasma membrane peptide transporters, PHT1 (and PHT2) is known for its capacity to transport histidine. However, the binding of histidine or histidine containing peptides was not observed in our assay. This may be the result of an intrinsic limitation of the thermal shift assay, where although binding occurs, this is not translated in a thermal shift. This has been observed for fragment screening campaigns, using different biophysical methods, where many of the positive hits were not detectable in thermal shift assays[52,53]. Therefore, other binding and transport assays are required before any further conclusions can be drawn regarding the role of histidine or histidine containing peptides in transport.

Besides characterizing PHT1 on a functional level, we elucidated the molecular basis for the interaction of PHT1 with the adaptor protein TASL and the structure of PHT1 in the outward-open conformation. In this state, PHT1 cannot recruit and bind the TASL adaptor. The AF2 model of the PHT1-TASL complex was extensively verified by mutational studies in vitro and in vivo. The first 16 residues of TASL form a tight interaction network with PHT1, in agreement with previous data suggesting that the first 8 residues are crucial for PHT1-TASL binding[29]. The 16 residues of TASL are part of an α-helix that extends from the PHT1 inward-open central cavity down to the cytosol. For a natural substrate of PHT1, the di- or tripeptide is clamped by the highly conserved Arg44 and Lys180 at the C-terminus and Glu473 at the N-terminus. The TASL Glu4 side chain mimics the C-terminus of a natural substrate, thus is involved in a salt bridge with Arg44 and Lys180 of PHT1, while the free amino group at the first methionine residue of TASL is coordinated by Glu473. The N-terminal coordination of TASL is critical for the interaction, confirmed by the mutational studies. Since TASL binds in the central pocket of PHT1 with high affinity, we expect that any transport activity is blocked.

In fact, TASL knockdown experiments in monocytes showed an increased pH level of the lysosomal lumen, while overexpression of TASL promoted its acidification[54]. These observations indicate that TASL may regulate the lysosomal environment by inhibiting the transport activity of PHT1, hence, the local concentration of protons inside the lysosome is increased. This hypothesis is in line with PHT1 role in regulating the lysosomal pH[26] and its interaction with TASL.

The critical function of PHT1 opens a path for the development of inhibitors with anti-inflammatory activity. Efforts in this area have already started, with the identification of the compound AJ2–30, that can effectively block PHT1[55]. Indeed, AJ2–30 was shown to inhibit MDP transport, and resulted in the disruption of TLR7-9-mediated mTOR activation and IRF5/7 nuclear translocation. Yet, this compound did not affect TASL binding, and its mode of action in respect to PHT1 inhibition is still not clear. From a pharmaceutical point of view, a potent inhibitor that could block both, transport activity and TASL binding would be advantageous. One way to achieve this is the development of competitive inhibitors, that target the central binding pocket of the inward-open conformation or non-competitive inhibitors that can lock PHT1 in the outward-open conformation (just like Sb27). As seen for other mammalian POTs[33], the major structural rearrangements during transport occur in the N-domain, while the C-domain remains almost rigid. Given the role of the N-domain in conformational transitions, a good target area for selecting conformation specific binders would be the cytosolic loops of the N-domain. Lastly, the design of non-competitive exofacial inhibitors that can bind the central binding pocket in the outward-open conformation of PHT1. Such inhibitors could have several attractive advantages like the design of pH-dependent pro-drugs and enhanced selectivity[56], since they need to be targeted and delivered to the lysosome. The outward-open PHT1 structure determined in this study, is the ideal starting point for designs and structural investigations of such inhibitors.

Based on our findings, we suggest a model for the role and function of the peptide transporter PHT1 (Fig. 5). The expression of the adaptor protein TASL is restricted to the haemotopoietic compartment (specially to myeloid cells, B lymphocytes and plasmacytoid dendritic cells), while PHT1 has a much broader tissue distribution[29]. On the one hand, PHT1 can function as a proton coupled peptide transporter directly influencing the lysosome environment[24] and internalize NOD1/2 ligands in the cytoplasm enabling NOD signaling[17,25–27]. On the other hand, PHT1 has a transport activity-independent function. In endolysosomal TLR signaling, PHT1 is required as a receptor for the engagement of TASL and consequent IRF5 activation. More importantly, TASL can only bind the inward-open conformation of PHT1. This conformation dependent binding opens the question of a potential regulation mechanism to indirectly control the production of type I IFN genes. During the revision of this paper, a study by Zhang et al.[57] was published highlighting that PHT1 transport activity is not necessary for IRF5 activation and cytokine production, when TASL is tethered to the endolysosome. This study further validates our model and agrees with our data. Future work is necessary to understand what triggers TASL engagement with PHT1 and how this activation is linked to TLR ligand recognition and signaling responses.

Our work sheds light on the molecular determinants for the recruitment of TASL by PHT1 and its influence on the transport activity. Understanding how TASL binds PHT1 will accelerate the development of drugs that can effectively disrupt TASL-IRF5-mediated INF-I production. At the same time, our experimental structure forms the basis for the development of exofacial or non-competitive inhibitors, which can trap PHT1 in the outward-open conformation and block both transport activity and TASL binding.

## Methods

### Construct design, expression and purification of PHT1

The gene encoding full-length chicken PHT1 (Uniprot accession number F1NG54), was inserted into an pXLG expression vector[58] with an N-terminal Twin-Streptavidin tag followed by the human rhinovirus 3C (HRV-3C) protease recognition sequence and a C-terminal Avi-tag for enzymatic biotinylation. To target PHT1 to the plasma membrane, two point mutations L12A and L13A, were inserted at the N-terminus of PHT1 (PHT1 L12A/L13A). All PHT1 mutants were generated with a Standard PCR-based strategy. Based on previously protocols[59], PHT1 was transiently transfected in Expi293F (Thermo Fisher Scientific, Cat.# 100044202) or Expi293F GnT1⁻ (Thermo Fisher Scientific, Cat.# A39240) cells, at a final density of $20 \times 10^6$ cells per ml. DNA was added to a final concentration of 1 mg per l, mixed and followed by addition of MAX PEI (Polysciences) in a 1:3 ratio (w/w). For protein expression, transfected cells were cultured in FreeStyle293 Expression Medium (Thermo Fisher Scientific) at a cell density of $1 \times 10^6$ cells per ml and

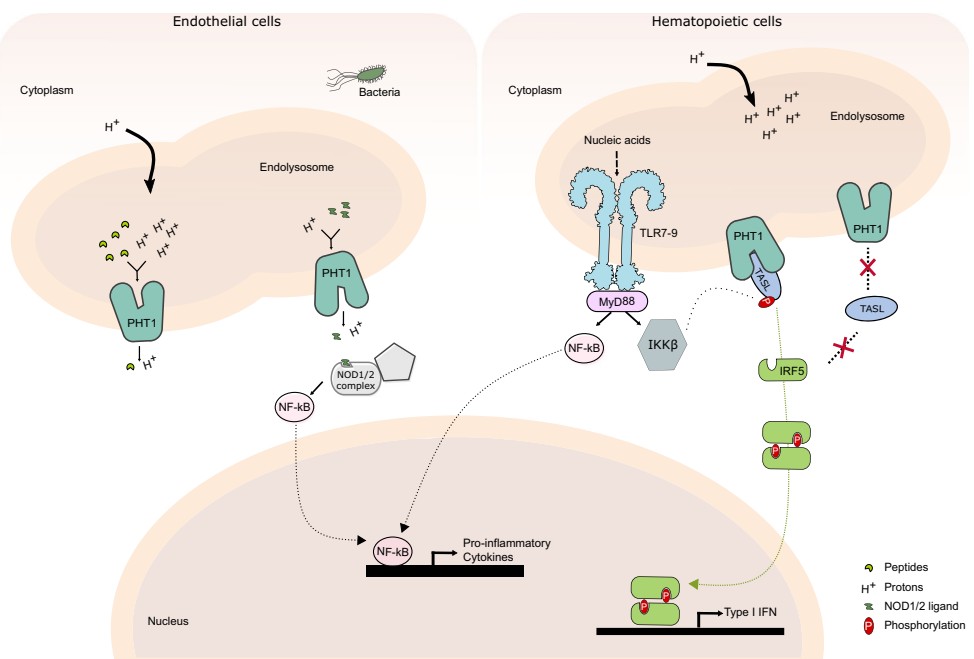

**Fig. 5 | Proposed mechanism for PHT1 mode of action.** In cell types not expressing TASL, PHT1 uses the proton motive force for the transport of oligopeptides and other ligands from the lumen of the endolysosome to the cytosol, thereby regulating endosomal pH but also enabling NOD1/2 complex formation leading to NF-κB activation. In endolysosomal TLR signaling, PHT1 has a transport activity-independent function. PHT1 acts as a receptor for the engagement of TASL. Once the complex is formed, the adaptor protein TASL can be phosphorylated by the IKKβ kinase and in turn can recruit and activate IRF5 and subsequently promote the production of type I IFN genes. TASL can only recognize the inward-open conformation of PHT1. If the transporter is locked in the outward-open conformation, the signaling pathway will be inhibited.

incubated at 37 °C with agitation at 220 rpm in an 8% CO₂ atmosphere for 3 days. Cells were collected by centrifugation and resuspended in 50 mM Tris pH 7.5, 300 mM NaCl, 2 mM MgCl₂ and 5% glycerol supplemented with cOmplete EDTA-free protease inhibitors (Roche) and DNaseI (Sigma Aldrich). Whole cells were solubilized in 1% N-dodecyl-β-D-maltopyranoside (DDM, Anatrace) and 0.2% cholesteryl hemisuccinate (CHS, Anatrace) for 2 h at 4 degrees under mild agitation. Insoluble material was removed by ultracentrifugation at 100,000 × *g* for 30 min. 5 mM EDTA was added to the cleared lysate followed by incubation with Strep-Tactin beads (IBA) for 30 min at 4 °C on a rotating wheel. The resin was washed with 5 column volumes of a buffer containing 50 mM Tris pH 7.5, 300 mM NaCl, 5% glycerol, 1 mM EDTA and 0.1% DDM/ 0.02% CHS, followed by 5 column volumes of the same buffer containing only 0.02% DDM/ 0.004% CHS. The protein was eluted by a buffer containing 50 mM Tris pH 7.5, 300 mM NaCl, 5% glycerol, 1 mM EDTA and 0.02% DDM/ 0.004% CHS and 10 mM desthiobiotin (IBA). The eluate was concentrated using an Amicon Ultra centrifugal filter (100-kDa molecular-weight (MW) cut-off, Millipore), and further purified by gel filtration chromatography (Superdex 200 increase 10/300 GL, GE Healthcare) in 50 mM HEPES pH 7.5, 200 mM NaCl and 0.02% DDM/0.004%CHS.

### Sybody selections, expression and purification
For sybody selections, enzymatic site-specific biotinylation of the Avi-tag was carried out before the last purification step. Purified PHT1 L12A/L13A, expressed in ExpiGnT1⁻ cells, was incubated at 4 °C overnight with purified BirA in 50 mM HEPES pH 7.5, 200 mM NaCl, 0.02% DDM/0.004%CHS, 10 mM magnesium acetate, 5 mM ATP and a two-fold excess of biotin over the Avi-tag containing PHT1 sample concentration. Purified 3C protease and EndoH enzymes in a 1:10 fold molar concentration over PHT1 were added to the same reaction to remove the Twin-Streptavidin tag and to deglycosylate the sample, respectively. Next day, the mixture was incubated with Ni-NTA beads

(Invitrogen) to remove BirA, EndoH and the 3C protease followed by gel filtration chromatography, as described above.

The Sybody Generation Toolbox was a gift from Prof. Markus Seeger (University of Zurich) and selections were performed as described in detail[42,43], with one major difference. After one round of ribosome display and two rounds of phage display, we obtained enrichments against the target protein twofold above compared to the control protein (human PepT2). However, the enrichment values were quite low, especially for the concave library, with only a 2.2-fold enrichment compared to the control protein. To further enrich the pool of binders, a third phage display round was performed, combined with another off-rate selection, to obtain binders with increased affinities. During the second off-rate selection, 3 μM of non-biotinylated PHT1 with a 10 min incubation time was used. Unique ELISA positive clones were expressed and purified as described previously[60]. For large scale purifications, Sb27 inserted into the pSBinit plasmid was transformed into *Escherichia coli* MC1061 F⁻ (Lucigen) cells and expressed in TB medium overnight at 22 °C, after induction with 0.02% (w/v) L-Arabinose. The proteins in the periplasmic space were released by osmotic shock and purified by immobilized metal affinity chromatography (IMAC) using Ni-NTA resin (Invitrogen) in TBS buffer (20 mM Tris, 150 mM NaCl, pH 7.4). Protein was eluted in TBS buffer containing 300 mM Imidazole and concentrated using an Amicon Ultra centrifugal filter (10-kDa molecular-weight (MW) cut-off, Millipore). Sb27 was further purified by gel filtration chromatography (Superdex 75 10/300 GL, GE Healthcare) in TBS buffer.

### Cryo-EM sample preparation and data collection of PHT1 in complex with Sb27
For Cryo-EM, Strep-Tactin purified PHT1 L12A/L13A expressed in ExpiGnT1⁻ cells, was incubated with a two-fold molar excess of Sb27 and EndoH enzyme in a 1:10 fold molar concentration over PHT1, overnight at 4 °C. The sample was further purified by gel filtration

chromatography using a Superdex 200 increase 10/300 GL column, to separate EndoH and excess of Sb27, as described above. Just before sample vitrification, the complex was further subjected to a final gel filtration chromatography step (Superdex 200 increase 5/150 GL, GE Healthcare) in 10 mM MOPS, pH 7.0, 100 mM NaCl, 0.01% DDM/ 0.002% CHS. The peak fraction reached a concentration of 53 μM, and 3.5 μl of it was applied to a freshly glow-discharged gold holey Ultrafoil 1.2/1.3 300 mesh grids (Quantifoil). After 3 s of blotting in 100% humidity at 4 °C, the grid was plunged into liquid propane using a Mark IV Vitrobot (Thermo Fisher Scientific). Cryo-EM data collections were performed by using a Titan Krios (Thermo Fisher Scientific) microscope (Thermo Fisher Scientific), running at 300 kV equipped with a Gatan K3 camera in the electron-counting mode and BioQuantum energy filter (Gatan) set to 10 eV. Data collection settings are summarized in Supplementary Table 2. Imaging was performed at a nominal magnification of ×105,000 and a physical pixel size of 0.85 Å, with a 70-μm C2 aperture and 100-μm objective aperture at a dose rate of 19.5 e$^-$/pixel per second. The images were collected as exposures of 2.75 s spread in 50 frames, resulting in a total dose of 75 e$^-$/Å$^2$ per movie. Data was collected using EPU2.8.0.1256REL (Thermo Fisher Scientific).

### Data processing, model building and refinement

Images from two independent datasets collected with the same optics settings, were analyzed, side by side, using Relion-3.1[61] and CryoSPARCv3[62] software. The data-processing workflow is summarized in Supplementary Fig. 4c. In both datasets, raw movies were motion-corrected using MotionCor2[63] in Relion. Contrast transfer function were determined on dose weighted micrographs using CTFFIND4[64]. Particles were extracted using CrYOLO[65], with a 200-pixel box and binning to 50 pixels, and were subjected to multiple rounds of 2D classification in Relion-3.1. The best representative classes were selected manually. These particles served as input for ab initio reconstructions in CryoSPARCv3, which were used as references for 3D classification refinement in Relion-3.1. Particle trajectories and cumulative beam damage were further corrected for, in Relion-3.1 using the Bayesian polishing approach. Shiny particles were imported in cryoSPARCv3 for further analysis. Ab initio volumes were reconstructed from the imported particles, and used for rounds of particle sorting by heterogeneous refinement. Selected images were then used for non-uniform refinement, resulting in a 3.7 Å reconstruction for the first dataset, and a 3.5 Å for the second one, which looked identical after careful visual inspection in ChimeraX[66] (no conformational heterogeneity observed). The particles used for these two reconstructions were therefore merged, and subjected to a round of heterogeneous refinement using a "good" and a "bad" volume. The set of projections clustered in the "good" class (1,328,233 particles) was finally used for a final non-uniform refinement resulting in a 3D reconstruction with a global resolution of 3.3 Å, as estimated by gold-standard Fourier shell correlation (FSC) using a 0.143 cutoff. Post processing in DeepEMhancer[67] using the two half maps as input and the default tight target model resulted in a more interpretable map for the Sb27 molecules, while the model was build and refined in a map sharpened in CryoSPARCv3. The chicken PHT1 and Sb27 models were predicted by AlphaFold[47] and docked into the PHT1-Sb27 map in chimera[68]. Two molecules of Sb27 could be readily fitted into the map. The models were then subjected to several iterative cycles of manual model adjustment in Coot 0.9.8.1[69] and Isolde[70] and real-space refinement using Phenix[71]. In PHT1, a stretch of the bundle bridge (residue 279–300) and two loops at the lumen side (residues 135–159 and 362–370) connecting TM3/TM4 and TM7/TM8, respectively, were not visible in the map and were excluded from model building in Coot. Geometry was validated using MolProbity 4.2[68] in Phenix. Structure figures were prepared in ChimeraX 1.3[66] or PyMOL (Molecular Graphics System,

v1.5.0.4 (Schrödinger)). Conservation of residues across species was analyzed using ConSurf[72]. Sequence alignments were performed in PROMALS3D[73] and visualized using ALINE 1.0.025[74]. The phylogenetic tree was constructed in MEGA v10.1.8[75].

### Structural modeling of the chicken PHT1-TASL complex

To model the chicken PHT1-TASL complex, we used the 'custom' mode of AlphaPulldown[49], a Python package built upon AlphaFold[47] and AlphaFold Multimer version 2.3.0[48]. All parameters were set to default except for the max_recycles, which was increased from 3 to 12 for better model quality and to avoid steric clashes. The local quality of the model was assessed by predicted local distance difference test (pLDDT) scores as returned by AlphaFold. The confidence in the relative arrangement between PHT1 and TASL proteins was evaluated by predicted aligned errors (PAE), also as returned by AlphaFold. Other evaluations of the model quality and properties, including Predicted DockQ score (pDockQ)[76], protein-interface score (PI-score)[77], and biophysical properties of the interaction interface (using PI-score program[77]), were calculated by the AlphaPulldown package.

### Thermal stability assays

The thermal stability of PHT1 or PHT1-Sb27 was monitored by nanoDSF[38,39] at a concentration of 0.4 mg/mL in gel filtration buffer (50 mM HEPES, pH 7.5, 100 mM NaCl, 0.02% DDM/ 0.004%CHS). For the ligand screen, PHT1 at 0.4 mg/ml was incubated with 5 mM of the different compounds. The pH screen, was performed as described before, in either 50 mM HEPES, pH7.5, 100 mM NaCl, 0.02% DDM/ 0.004%CHS, or 50 mM MES, pH 6.5, 100 mM NaCl, 0.02% DDM/ 0.004%CHS, or 50 mM MES, pH 5.5, 100 mM NaCl, 0.02% DDM/ 0.004%CHS. To monitor binding induced thermal shift changes of PHT1 with TASL peptides, PHT1 at 0.4 mg/ml was incubated with 50 μM peptides in gel filtration buffer containing 5 % DMSO. TASL peptide titrations were performed in a similar manner, by incubating PHT1 with eleven 2-fold peptide dilutions, with a starting concentration of 100 μM. PepT1 and PepT2 proteins were purified as described in[33], and binding assays were performed in a similar manner as described for PHT1. PHT2 was purified as described previously for PHT1, and binding assays were performed in a similar manner as described for PHT1. All samples were incubated for at least 10 min at RT prior analysis. Standard grade nanoDSF capillaries (Nanotemper) were loaded into a Prometheus NT.48 device (Nanotemper) controlled by PR. ThermControl (version 2.1.2). Excitation power was adjusted to 30–50% and samples were heated from 20 °C to 90 °C with a slope of 1 °C/min. With the exception of TASL peptide titrations, all samples were run in triplicates and error bars represent standard deviations of three biological replicates. Data were analyzed using MoltenProt[78] and $K_D$'s were fitted following a modified Hall's method[79] for determining the affinity from irreversible thermal shits, described here[80]. All figures were prepared using Prism 9 (GraphPad).

### Biolayer interferometry (BLI)

The binding of Sb27 to PHT1 was measured by biolayer interferometry (BLI) using the Octet RED96 system (FortéBio). Biotinylated PHT1 expressed in ExpiGnT1- cells, was loaded on streptavidin biosensors (FortéBio), pre-equilibrated in 50 mM HEPES, pH 7.5, 100 mM NaCl, 0.02% DDM/ 0.004%CHS supplemented with 0.5% (w/v) BSA, for 20 min. Prior to association, a baseline step of 60 s was performed. Subsequently, sensors were dipped in increasing concentrations of Sb27 ranging from 6–800 nM for 8 min followed by a 12 min dissociation step. All experiments were carried out at 22 °C. Data were reference-subtracted and aligned to each other in the Octet Data Analysis software v10.0 (FortéBio), using a 1:1 binding model. All figures were prepared using Prism 9 (GraphPad).

## Microscale Thermophoresis (MST)

The binding affinity of PHT1 to $TASL_{1-13}$ was measured by microscale thermophoresis using the MST NT.115 device (NanoTemper Technologies). A Cy5 dye was fused to the C-terminus of the $TASL_{1-13}$ peptide containing a polyethylene glycol (PEG) spacer. A 2-fold dilution series of PHT1 protein was prepared in 50 mM HEPES, pH 7.5, 100 mM NaCl, 0.02% DDM/ 0.004%CHS and 5 % DMSO with the highest concentration in the assay being 28 μM. Labeled $TASL_{1-13}$ peptide at a concentration of 25 nM was mixed with the substrate dilution series and incubated for 10 min at room temperature before loading the samples in standard capillaries. For the competition assay, the same procedure was used but 120 μM of Sb27 or negative control sybody were added to each PHT1 dilution series and incubated for 10 min prior to mixing with labeled TASL peptide. Measurements were performed using 20 % LED and medium MST power. Three independent experiments were performed, yielding similar results. The data were analyzed using the MO. Affinity Analysis software (NanoTemper Technologies). Figures were prepared using Prism 9 (GraphPad).

## Pull-down assays

The genes encoding the full-length chicken PHT1 (Uniprot accession number F1NG54), full-length human PHT1(accession number Q8N697), full-length PHT2 (accession number Q8IY34), full-length chicken TASL (Uniprot accession number A0A1L1RS25) and full-length human TASL (Uniprot accession number Q9HAI6) were inserted into an expression construct based on the pXLG vector[58]. For chicken PHT1, human PHT1 and human PHT2, a Twin-Streptavidin tag followed by the human rhinovirus 3C (HRV-3C) protease recognition sequence was introduced at the N-terminus, while for chicken and human TASL a C-terminal TEV-cleavable His/GFP fusion tag was inserted. The full-length human PepT2 (Uniprot accession number Q16348) and full-length human PepT1 (accession number P46059) were introduced into the pXLG vector, as described previously[33]. All mutations were introduced by PCR-based mutagenesis.

Wild-type or mutant PHT1 (or PHT2, or PepT1, or PepT2) and wild-type or mutant TASL were transiently co-transfected in Expi293F cells, in a 1:3 PHT1:TASL DNA ratio and to a final concentration of 1.5 mg per l, mixed and followed by addition of PEI Max (Polysciences) in an 1:2 ratio (w/w). Transfected cells were cultured for 2 days as described above. Cells were collected and solubilized in an identical manner as performed for the PHT1 purification. A 5 μl aliquot of cleared lysates was subjected to a western blotting assay using an anti-GFP-HRP antibody (Milteny Biotec) to determine expression levels of TASL wild-type and mutants. The remaining lysate was added to Strep-Tactin beads and incubated for 20 min at 4 °C. The beads were washed three times with 0.8 ml of a buffer containing 50 mM HEPES pH 7.5, 200 mM NaCl, 0.05% DDM/0.01%CHS, 1 mM EDTA and proteins were eluted in the same buffer containing 10 mM desthiobiotin. A 10-μl aliquot of the eluate was subjected to a western blotting assay using an anti-GFP-HRP antibody (Miltenyi Biotec, catalog number 130-091-833) with a 1:4000 dilution for TASL detection or using Strep-tactin-HRP (IBA, catalog number 2-1502-001) with a 1:4000 dilution for PHT1 detection. The chemiluminescent signal was detected by SuperSignalWest Pico PLUS (Thermo Scientific). Uncropped and unprocessed scans of western blots are provided in the Source Data file.

## Reporting summary

Further information on research design is available in the Nature Portfolio Reporting Summary linked to this article.

## Data availability

The EM data and fitted models for chicken PHT1 have been deposited in the Electron Microscopy Data Bank under accession code EMD-16758 and the PDB under accession code 8CNI. The outward-open structure of human PepT1, used for comparative analysis in this study, can be found in the PDB under accession code 7PMX. The AlphaFold model of the complex has been deposited in the ModelArchive [https://doi.org/10.5452/ma-oj2xo]. All protein sequences used in this study are publicly available at Uniprot (https://www.uniprot.org/) with the following accession codes: chicken PHT1 (Uniprot accession number F1NG54), human PHT1 (accession number Q8N697), PHT2 (accession number Q8IY34), chicken TASL (Uniprot accession number A0A1L1RS25), human TASL (Uniprot accession number Q9HAI6), human SLC15A5 (accession number A6NIM6), human Pept1 (accession number P46059) and human PepT2 (accession number Q16348). Source data are provided with this paper.

## Material availability

All reagents generated in this study are available from the Lead Contact with a completed Materials Transfer Agreement.

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

## Acknowledgements

We thank the Sample Preparation and Characterization facility at EMBL (Hamburg, Germany) and the team of the cryo-EM Facility at CSSB for their support, technical assistance and advice. We acknowledge Katharina Jungnickel for initial expression screening of PHT1, Nikolay Dobrev and all group members for fruitful discussions and continuous support and feedback on the project. This study was supported by a BMBF grant (number: 05K18YEA) to C.L.. Part of this work was performed at the CryoEM Facility at CSSB, supported by the UHH and DFG (grant numbers INST 152/772-1|152/774-1|152/775-1|152/776-1|152/777-1 FUGG). M.K. and D.Y. are part of a joint PhD degree program between EMBL, and Heidelberg University, Faculty of Biosciences.

## Author contributions

Conceptualization: T.F.C. and C.L. Methodology: T.F.C., M.K., D.Y., V.P., D.T., A.P., G.S., M.G., J.K. and C.L. Investigation: T.F.C., M.K., D.Y., V.P., D.T. and C.L. Visualization: T.F.C., M.K. and D.Y. Funding acquisition: A.P., M.G., J.K. and C.L. Project administration: C.L. Writing—original draft: T.F.C. and C.L. Writing—review and editing: all authors

## Funding

## Competing interests

The authors declare no competing interests.
