## [Peer Review File · Nature Communications]

REVIEWER COMMENTS

Reviewer #1 (Remarks to the Author):

This is the first structure of an intracellular localised oligopeptide transporter. The structure of the lysosomal peptide transporter SLC15A2 (PHT1) is very interesting and, in particular, the interaction with TASL. The Cryo EM structure is very solid and I would like to have seen this information used to help answer some more fundamental questions between the differences between PHT1 and plasma membrane localised PepT1 and PepT2 isoforms. The authors have rather focused on the interaction with TASL, yet the interpretation of thermal-shift assays and pull-down experiments is based on the AlphaFold model prediction of the PHT1-TASL interaction instead. More worryingly, we still have no idea of the mechanism for recruitment of TASL to PHT1. Surely, TASL can't bind to the inward state of PHT1 indiscriminately as then they transporter could no longer transport. I am positive about the study, but in its current form it's a bit too preliminary and hasn't been able to provide any deeper mechanistic insights. I have some suggestions and comments to this effect below:

The functional characterisation of PHT1

. The authors mention that there has been some conflicting reports about the substrate specificity in PHT1 based on cell based assays and the need for in vitro validation. It would be helpful to explain in more detail what the conflicting results are.

- The analysis in substrate preferences here is based on thermal-shift assays (TSA). While I have no issue with using TSA as an initial screening approach, it does have its limitations in extracting deeper insights. From the presented data we learn that both di and tri peptides can bind to PHT1. Given how "amazingly" similar the structure of the substrate binding site is between PHT1 and the outward-open AF-bound structure of human PepT1, this makes sense. The elephant-in-the-room is why is PHT1 needed for lysosomes when PepT1 is so similar? The lysosomes have a low pH (around 4) and one would expect that PHT1 would have evolved to work under these more acidic environments. For example, one might expect that it can bind neutral forms of D and E amino acids or the protonated state of histidine in di and tripeptides. Yet, in the thermal shift assays we see that all histidine containing di-peptides and glutamate or aspartate di-peptides are actually poorly binding. Indeed, the stronger binding is seen for lysine and arginine containing di-peptides, yet these residues will be protonated all they up to pH 9. Basically, I can't make any sense of the thermal shift assays. Could it be a problem that binding was carried out at pH 7.5? In the cell di and tri-peptides are binding in the lysosomal pH 4 and then "releasing" the peptides into the cytoplasm at pH 7.2? The TSA further needs some benchmarking. For example, it's possible that a drop in a melting temperature of 2C means that the substrate is no longer transported, even though it still can bind. It would be helpful to show side-by-side comparisons

between PHT1 and PepT1 in the TSA results (of a select few) to get a better impression of the differences.

.- We desperately need a proteoliposome transport assay to make sense of the TSA results. I am sure the authors have tried. Given that PepT1 and PepT2 work so well in proteoliposome transport assays and given PepT1 is so similar in structure to PHT1, it does make you wonder why its not working if thats the case? Is there differences outside of the substrate binding pocket the might explain a preference to more acidic environments and challenges with establishing in vitro transport assays? Even if proton-coupled experiments are difficult to get to work because of the pH asymmetry requirements, I would expect that counter-flow experiments should work. Along these lines, is there any indication from the structure as to how PHT1 is proton coupled? Is there a histidine residue in a similar position to PepT1 for example?

The complex interaction of PHT1 TASL

Pull-down assays of PHT1 with co-expressed TASL mutants and thermal shift assays between purified PHT1 and TASL peptides is very well done. I think its clear that TASL binds to PHT1 and what features are important for the interaction. I am not sure what to make of these detailed interaction figures based on a AlphaFold model though. Its possible that is correct, but the element of doubt means that it is only a model ...

- Please show that TASL doesn't bind to PHT1 in the presence of the slybody Sb27b, which stabilises and outward-open state. Given the high affinity of the peptide interaction I should ITC should be possible here.

- Confirming it binds to the inward-open state how do you know that the first region of the TASL peptide mimics a natural substrate (Fig. 4a)? I couldn't find the di or tripeptide binding data matching the TASL sequence in Fig. 1b. Also, based on the model a glutamate in the fourth position seems critical to interact with the proposed positively charge residues, yet in Fig. 1b now of the peptides containing a glutamate actually bind. Again it would be helpful to have a transport assay to measure competition of TASL to actual transport.

- I noticed that there was no interaction with TASL and PepT2 in the pull down assays based on co-expression, How about with purified components? Does TASL bind to PepT1?

Overall, even if the authors are able to better validate the interaction with TASL and PHT1 we still have NO mechanism. I cant be that TASL binds to PHT1 in an inward open conformation "every time" as this

would mean that PHT1 could never go through a transport cycle and move peptides out of the lysosome. So what triggers TASL to bind in a physiological context?

Reviewer #2 (Remarks to the Author):

The manuscript titled "Molecular basis of TASL recruitment by PHT1" by Custodio and Loew et al reports structural and functional characterization of PHT1, a peptide transporter, and establishes a structural model of the TASL-PHT1 complex by validating computational models of the complex with binding and mutational analyses. The structure of a mammalian homolog of PHT1 in complex with a sybody was determined by cryoEM to 3.3 Angstrom resolution, which is very good for a transporter of this size. The structure enhances our knowledge of the POT or SLC15 family of transporters. The structure also provides a template for understanding interactions between PHT1 and TASL. Functional analysis of PHT1 relies mainly on substrate binding, and although the outcome is solid and interesting in terms of demonstration of substrate selectivity, there is a missed opportunity here to understand the rate of substrate transport or the stoichiometry of co-transported protons.

Perhaps the main imperfection of the study is a lack of a TASL-PHT1 structure determined by cryoEM. Nevertheless, the authors did an extensive study of the interactions by first producing computational models of the complex, and then examining the models by mutational studies. A self-consistent picture emerged from the study, and it is also consistent with published prior knowledge of the complex.

Overall I remain positive about the study and feel that the authors are rigorous in their experiments and cautious in data interpretations. I would like to raise the question on whether TASL has to assume an alpha helix conformation in order to satisfy the mutational data. Could the peptide binding pocket in PHT1 simply accommodate or recognize a non-structured tri or quatro peptide from TASL?

Response to Reviewers Comments:

Reviewer #1 (Remarks to the Author):

This is the first structure of an intracellular localised oligopeptide transporter. The structure of the lysosomal peptide transporter SLC15A2 (PHT1) is very interesting and, in particular, the interaction with TASL. The Cryo EM structure is very solid and I would liked to have seen this information used to help answer some more fundamental questions between the differences between PHT1 and plasma membrane localised PepT1 and PepT2 isoforms. The authors have rather focused on the interaction with TASL, yet the interpretation of thermal-shift assays and pull-down experiments is based on the AlphaFold model prediction of the PHT1-TASL interaction instead. More worryingly, we still have no idea of the mechanism for recruitment of TASL too PHT1 Surely, TASL cant bind to the inward state of PHT1 indiscriminately as then they transporter could no longer transport. I am positive about the study, but in its current form its a bit too preliminary and hasn't been able to provide any deeper mechanistic insights. I have some suggestions and comments to this effect below:

Response: We thank the reviewer for the support of our work and we have tried to strengthen our manuscript by compelling with several of the suggestions and comments.

The functional characterisation of PHT1

The authors mention that there has been some conflicting reports about the substrate specificity in PHT1 based on cell based assays and the need for in vitro validation. It would be helpful to explain in more detail what the conflicting results are.

Response: This was indeed unclear, and we have adjusted this.

It now reads: "Different cell-based assays were used to monitor uptake, which could be the reason for several inconsistencies found among published work ^{16,18,37}. For example, the first study on substrate selectivity of PHT1 was performed in *Xenopus Laevis* oocytes and the uptake of histidine ($K_m = 17 \mu\text{M}$) and carnosine with maximum activity at pH 5.5 could be detected. However, the subcellular location of PHT1 was not considered and its expression at the plasma membrane was not confirmed ³⁷. Another study in 2005 by Bhardwaj *et al.*, ¹⁸ used the transient transfection method for the expression of PHT1 in COS-7 cells, and the authors described the uptake of histidine and carnosine but not of glycylsarcosine (GlySar), a known PepT1 substrate. Again, the authors did not show the expression of PHT1 at the plasma membrane. In a more recent study ¹⁶, a plasma membrane mutant of PHT1 was stably transfected in MDCK cells and the uptake of Histidine, carnosine and also GlySar was reported. The substrate uptake was pH dependent with a maximum activity at pH 6.5. Furthermore, histidine inhibition studies from the same study yielded an IC_{50} value for histidine greater than 1 mM, considerably different from the reported K_m value. The observed variations between the published studies might be explained by different expression systems or different type of methodology used (i.e., transient vs stable transfections) and arguably the presence of other endogenous transport systems."

- The analysis in substrate preferences here is based on thermal-shift assays (TSA). While I have no issue with using TSA as an initial screening approach, it does have its limitations in extracting deeper insights. From the presented data we learn that both di and tri peptides can bind to PHT1. Given how “amazingly” similar the structure of the substrate binding site is between PHT1 and the outward-open AF-bound structure of human PepT1, this makes sense. The elephant-in-the-room is why is PHT1 needed for lysosomes when PepT1 is so similar? The lysosomes have a low pH (around 4) and one would expect that PHT1 would have evolved to work under these more acidic environments. For example, one might expect that it can bind neutral forms of D and E amino acids or the protonated state of histidine in di and tripeptides. Yet, in the thermal shift assays we see that all histidine containing di-peptides and glutamate or aspartate di-peptides are actually poorly binding. Indeed, the stronger binding is seen for lysine and arginine contacting di-peptides, yet these residues will be protonated all they up to pH 9. Basically, I cant make any sense of the thermal shift assays. Could it be a problem that binding was carried out at pH 7.5? In the cell di and tri-peptides are binding in the lysosomal pH 4 and then “releasing” the peptides into the cytoplasm at pH 7.2? The TSA further needs some benchmarking. For example, its possible that a drop in a melting temperature of 2C means that the substate is no longer transported, even though it still can bind. It would be helpful to show show side-byside comparisons between PHT1 and PepT1 in the TSA results (of a select few) to get a better impression of the differences.

Response: We agree on the statements made by the reviewer. We have now generated additional data and added them as part of supplementary figure 3. In supplementary figure 3, you can now find the TSA data for 65 di- and tripeptides screened against PHT1, PepT1 and PepT2 at two different pH values (pH 7.5 and 5.5).

In line 182, you can now read: “To better understand the substrate specificity/preference within the POT family, we have screened 65 di- and tripeptides against PHT1, PepT1 and PepT2 at two different pH values (Supplementary Figure 3). The plasma membrane transporters are stabilized by a broader set of peptides than PHT1. PepT2 as a known high-affinity and low-capacity transporter shows higher stabilization effects upon peptide binding, compared to PepT1 or PHT1. For all three tested transporters, peptides that stabilize the protein at pH 7.5, also stabilize it at pH 5.5, with the exception of a few dipeptides containing positive residues, like Lysine-Alanine (KA), Lysine-Proline (KP), Lysine-Lysine (KK) and Arginine-Proline (RP). This indicates that a protonation event in one or more residues that are part of the binding site of these transporters negatively influences binding of these positively charged peptides. Overall, peptides with negatively charged residues do not strongly stabilize any of the transporters. In conclusion, the substrate preference of PHT1 seems to be more restricted compared to PepT1 or PepT2.”

In general, we want to state that peptide binding does not automatically translate into transport. Furthermore, we are not able to make any statement/conclusions on peptides inducing negative thermal shifts.

- We desperately need a proteoliposome transport assay to make sense of the TSA results. I am sure the authors have tried. Given that PepT1 and PepT2 work so well in proteoliposome transport assays and given PepT1 is so similar in structure to PHT1, it does make you wonder why its not working if thats the case? Is there differences outside of the substrate binding pocket the might explain a preference to more acidic environments and challenges with establishing in vitro

transport assays? Even if proton-coupled experiments are difficult to get to work because of the pH asymmetry requirements, I would expect that counter-flow experiments should work. Along these lines, is there any indication from the structure as to how PHT1 is proton coupled? Is there a histidine residue in a similar position to PepT1 for example?

Response: Proteoliposome assays are indeed a highly valuable tool for transporter research. Liposome uptake assays for bacterial POTs are very well established, also in our lab. However, this is not the case for eukaryotic POTs. To our knowledge, there are only limited published data on proteoliposome uptake assays for eukaryotic POTs (Parker *et al.*, 2021). All other studies, have performed uptake assays using cell-based systems. Current problems are likely linked to both mammalian protein specific lipid dependencies, the asymmetry of the membrane and the challenges of creating tight proteoliposomes for a meaningful proton gradient. Since POTs are proton driven, the quality requirements for proteoliposomes are high as protons readily pass through leaky membranes. Furthermore, lysosomal membranes have very unique lipid composition which brings an extra challenge in case of PHT1.

We agree that uptake assays are needed to further explain the substrate selectivity of PHT1 and given that we failed monitoring peptide uptake via PHT1 by proteoliposome assays, we turned to cellular uptake assays in order to address the raised points. Here, however, we also encountered several difficulties. For cellular uptake assays we used HEK cells, because they were previously used to study the substrate uptake of both Human PepT2 and Horse PepT1. To target PHT1 to the plasma membrane, we mutated the lysosomal targeting signal. First we monitored the uptake of the fluorescent reporter peptide β -AK-AMCA. This assay is well established for several (but not all) bacterial POTs, and also for Human PepT2 and PepT1. However, we did not see any uptake of β -AK-AMCA by PHT1 (see figure below). Varying substrate concentration, buffer composition or pH did not change the results.

Figure 1 - Uptake of β -Ala-Lys-AMCA by HsPepT2 (orange), ggPHT1 (L12A/L13A) (green) and empty plasmid (grey) expressing cells. The uptake was performed with increasing concentrations of substrate (0 – 250 μ M) and an incubation time of 15 min.

Since the AMCA moiety of the β -AK-AMCA peptide could be detrimental for substrate recognition in PHT1, we next used tritium labeled histidine or the AA dipeptide to monitor direct uptake into HEK cells. We observe uptake of both compounds to a certain extent, but this is also the case for HEK cells transfected with an empty plasmid only. This high background is likely caused by endogenous transport systems in our cell model. Due to the high background, and hence

expected low signal to noise ratio, we do not believe that any data derived from these type of experiments would be conclusive and convincing.

Figure 2 -Cell uptake assays using tritium labeled substrates. a) Ala-Ala peptide activity of HsPepT2, PHT1 L12A/L13A and empty plasmid in ExpiGnT1- cells. Cells were incubated with 500 μ M Ala-Ala containing a tracer amount of ³H-Ala-Ala and uptake was measured after 15 min. b) Histidine uptake of PHT1 L12A/L13A or empty plasmid in ExpiGnT1- cells. Cells were incubated with 1 mM Histidine containing a tracer amount of ³H-Histidine and transport was measured after several time points (5, 15, 30, 60 and 90 min).

In conclusion, for the time being we cannot provide a convincing transport assay for PHT1. We agree that this type of assay is important to further understand the substrate selectivity by PHT1 and the differences among the members of the POT family. We will continue our efforts to establish a functional proteoliposome assay for mammalian transporters and also consider to implement an oocyte assay. Since we currently lack a functional transport assay, we focus in this manuscript on the characterization of the interaction of PHT1 with the immune adaptor protein TASL.

- Along these lines, is there any indication from the structure as to how PHT1 is proton coupled? Is there a histidine residue in a similar position to PepT1 for example?

Response: The histidine residue present in PepT1 and PepT2 that is known to be important for proton-coupled transport is not conserved in the other members of the POT family. PHT1 contains a Leucine residue in the equivalent position of the Histidine in PepT1 and PepT2, and no other buried histidine residues in the transmembrane region can be found. Nevertheless, several buried charged residues in the transmembrane region can be found, including residues from the conserved ExxER motif known to be important for proton coupled transport together with Lys180 as part of the substrate binding pocket but also Glu473 and Asp381 (similar to bacterial POTs).

The complex interaction of PHT1 TASL

Pull-down assays of PHT1 with co-expressed TASL mutants and thermal shift assays between purified PHT1 and TASL peptides is very well done. I think its clear that TASL binds to PHT1 and what features are important for the interaction. I am not sure what to make of these detailed interaction figures based on a AlphaFold model though. Its possible that is correct, but the element of doubt means that it is only a model ...

- Please show that TASL doesn't bind to PHT1 in the presence of the slybody Sb27b, which stabilises and outward-open state. Given the high affinity of the peptide interaction I should ITC should be possible here.

Response: We have addressed this point and added additional data as part of figure 3, panel f. We took advantage of the already established MST assays using the TASL1-13-Cy5 labeled peptide. Now we show that the TASL peptide only binds PHT1 in the absence of Sb27. The presence of an unrelated sybody (negative control) has no influence on binding.

You can now read: "Sb27 stabilized PHT1 in the outward-open conformation by forming an interaction network with the N-domain of PHT1. Cryo-EM reconstructions of a potential PHT1-Sb27-TASL1-13 complex resulted in the PHT1-Sb27 complex only, since access to the binding site for TASL was blocked. This has been further confirmed by monitoring binding of the fluorescently labelled TASL peptide to PHT1 in the presence of a constant concentration of Sb27 or an unrelated sybody, used as a negative control (SbNC). Our data show that the TASL peptide can only bind to PHT1 in the presence of SbNC but not in the presence of Sb27 (Figure 3f)."

- Confirming it binds to the inward-open state how do you know that the first region of the TASL peptide mimics a natural substrate (Fig. 4a)? I couldn't find the di or tripeptide binding data matching the TASL sequence in Fig. 1b. Also, based on the model a glutamate in the fourth position seems critical to interact with the proposed positively charge residues, yet in Fig. 1b now of the peptides containing a glutamate actually bind. Again it would be helpful to have a transport assay to measure competition of TASL to actual transport.

Response: With the current PHT1-TASL model at hand, we argue that the coordination of the first TASL residues in the binding site is similar to the coordination of di- and tripeptides known from bacterial and eukaryotic POT structures. The residues in PHT1 that likely coordinate a short peptide are the same that coordinate the first four residues of TASL. But here, the sidechain of Glu4 of TASL mimics the carboxy-terminus of a dipeptide, while the N-terminal amino group of TASL is coordinated by the strictly conserved Glu473 residue of PHT1, which also coordinates the amino-terminus of a dipeptide.

In line 396, you can now read: "The TASL Glu4 side chain mimics the C-terminus of a natural substrate, thus is involved in a salt bridge with Arg44 and Lys180 of PHT1, while the free amino group at the first methionine residue of TASL is coordinated by Glu473. The N-terminal coordination of TASL is critical for the interaction, confirmed by the mutational studies."

Despite not being able to measure competition of TASL within transport at this point, we show that TASL binding is competitive with a dipeptide for PHT1 (Supplementary Figure 12b and c).

- I noticed that there was no interaction with TASL and PepT2 in the pull down assays based on co-expression, How about with purified components? Does TASL bind to PepT1?

Response: We have added now additional data as part of Supplementary Figure 13, panel C. Pull-down assays highlight that TASL is only pulled down by Human or Chicken PHT1, but not by human PHT2, human PepT1 nor human PepT2. We did the same experiment with purified proteins, using the thermal shift assay with both chicken and human TASL peptides. Here, we

show that both chicken and human TASL peptides can stabilize human PHT1, but not human PHT2, human PepT1 nor human PepT2. The chicken PHT1 protein, is stabilized by the chicken TASL peptide but not by the human TASL peptide. In the pull-down assays, we can also observe that the human TASL protein is less efficiently pulled down by chicken PHT1 compared to human PHT1, indicating lower affinity between chicken PHT1 and the human TASL peptide. You can now read: “Lastly, we confirmed that TASL recruitment/binding is specific to PHT1 and not to other members of the POT family by both *in vitro* binding data of the TASL peptides and *in vivo* pull down assays with full-length TASL (Supplementary Figure 13c).”

- Overall, even if the authors are able to better validate the interaction with TASL and PHT1 we still have NO mechanism. I cant be that TASL binds to PHT1 in an inward open conformation “every time” as this would mean that PHT1 could never go through a transport cycle and move peptides out of the lysosome. So what triggers TASL to bind in a physiological context?

Response: PHT1 function as a TASL receptor for the activation of IRF5, does not exclude its function as a transporter. While TASL is restricted to the haematopoietic compartment, PHT1 has a very broad tissue distribution, including multiple cell types that do not express TASL.

Although, is still unclear how PHT1 controls the upstream TLRs signaling, in this study we propose that PHT1 transport activity is not involved in TASL recruitment, hence IRF5 activation and Type I IFNs production. Furthermore, we now understand how TASL binds to PHT1, and more importantly that binding is conformation dependent. Conformation dependent binding could be a regulation mechanism to indirectly control IRF5 downstream signaling by inhibiting TASL binding and consequent phosphorylation.

To better explain these notions, we have now added Figure 5 with a schematic summarizing the multifunctional role for PHT1.

In line 426, it now reads: “Based on our findings, we suggest a model for the role and function of the peptide transporter PHT1 (Figure 5). The expression of the adaptor protein TASL is restricted to the haematopoietic compartment (specially to myeloid cells, B lymphocytes and plasmacytoid dendritic cells), while PHT1 has a much broader tissue distribution²⁹. On the one hand, PHT1 can function as a proton coupled Histidine/peptide transporter directly influencing the lysosome environment²⁴ and internalize NOD1/2 ligands in the cytoplasm enabling NOD signaling^{17,25–27}. On the other hand, PHT1 has a transport activity-independent function. In endolysosomal TLR signaling, PHT1 is required as a receptor for the engagement of TASL and consequent IRF5 activation. More importantly, TASL can only bind the inward-open conformation of PHT1. This conformation dependent binding opens the question of a potential regulation mechanism to indirectly control the production of type I IFN genes.”

Reviewer #2 (Remarks to the Author):

The manuscript titled "Molecular basis of TASL recruitment by PHT1" by Custodio and Loew et al reports structural and functional characterization of PHT1, a peptide transporter, and establishes a structural model of the TASL-PHT1 complex by validating computational models of the complex with binding and mutational analyses. The structure of a mammalian homolog of PHT1 in complex with a sybody was determined by cryoEM to 3.3 Angstrom resolution, which is very good for a transporter of this size. The structure enhances our knowledge of the POT or SLC15 family of transporters. The structure also provides a template for understanding interactions between PHT1 and TASL. Functional analysis of PHT1 relies mainly on substrate binding, and although the outcome is solid and interesting in terms of demonstration of substrate selectivity, there is a missed opportunity here to understand the rate of substrate transport or the stoichiometry of co-transported protons.

Perhaps the main imperfection of the study is a lack of a TASL-PHT1 structure determined by cryoEM. Nevertheless, the authors did an extensive study of the interactions by first producing computational models of the complex, and then examining the models by mutational studies. A self-consistent picture emerged from the study, and it is also consistent with published prior knowledge of the complex.

Overall I remain positive about the study and feel that the authors are rigorous in their experiments and cautious in data interpretations.

Response: We thank the reviewer for the support of our work.

I would like to raise the question on whether TASL has to assume an alpha helix conformation in order to satisfy the mutational data. Could the peptide binding pocket in PHT1 simply accommodate or recognize a non-structured tri or quatro peptide from TASL?

Response: To determine a Cryo-EM structure of the PHT1-TASL complex we currently lack a fiducial marker that binds and stabilizes the transporter in the inward open-state allowing binding of TASL. We have already tried several strategies: (i) Structure determination in the absence of a fiducial marker and (ii) multiple engineering approaches to increase the particle size to improve particle alignment. However, all these approaches failed so far. But we believe that the presented PHT1-TASL AlphaFold model is highly reliable, since we validated all observed and relevant interactions of the model by biochemical and biophysical assays. Therefore, we are convinced that the 'helical insertion model' of the N-terminal TASL residues holds. We believe that a short (3-4 residues) peptide from TASL also interacts with PHT1, but with very low affinity, which would be hard to measure. Our current model indicates that multiple residues of the first TASL residues contribute to the interaction (and not only the first 3-4 residues) and a helical conformation is required.

We also tried CD spectroscopy to monitor potential helix formation of the TASL peptide upon binding to PHT1 (however the expected signal change/increase in overall helicity is less than 2%). We could confirm that the TASL peptide (residues 1-13) is unfolded in aqueous solution, but the data were not interpretable as soon as cholesterol hemisuccinate (required by PHT1) was present in the sample due to its high absorbance in the far-UV range.

Alternatively, one could use NMR spectroscopy on a labelled TASL peptide and monitor chemical shift changes upon PHT1 binding. However, such an experiment is currently out of scope for this study.

REVIEWERS' COMMENTS

Reviewer #1 (Remarks to the Author):

The authors have significantly improved the paper and I appreciate their additional efforts, in particular for establishing a transport assay for PHT1. The data supporting that TASL binds to PHT1 and not to other peptide transporters and that it binds to the inward-open conformation is convincing.

The outward-open state provides a basis to understand substrate specificity, but this seems to fall short. The main issue remains that the transporter is designated as a histidine transporter, yet the authors cannot detect binding of any histidine- or histidine-containing peptides in PHT1. The authors indicate this could be a limitation in the thermal-shift assay. While this may be possible, the di-histidine peptides shows the strongest stabilisation at of all the di-peptides to PepT2. To me, this indicates that histidine-contains peptides is not a limitation of the assay itself. Indeed, if it was a limitation, then it would somewhat undermine all the peptides screening carried out here.

The authors has tried to monitor uptake of histidine in PHT1 targeted to the plasma membrane, but due to background level being too high they cannot come to any firm conclusions. I am not sure what the other studies have done differently? As far as I am aware, I thought that PepT1/PepT2 were not capable of transporting free amino acids? Given the close structural similarity of PHT1, is there any indication that could enable PHT1 to transport free amino acids in addition to di-peptides?

From my point of view, it therefore seems inconsistent to have a mechanism of PHT1 based on histidine transport when this cannot be confirmed here. I would prefer that the authors are clearer in the introduction and mechanism figure. It might be more prudent to outline that there data does not support the binding of histidine or histidine-containing peptides and as, such, how PHT1 is activated and linked to TASL needs further validation.

Reviewer #2 (Remarks to the Author):

The authors responded to my comments. I have no further comments.

Response to Reviewers Comments:

Reviewer #1 (Remarks to the Author):

The authors have significantly improved the paper and I appreciate their additional efforts, in particular for establishing a transport assay for PHT1. The data supporting that TASL binds to PHT1 and not to other peptide transporters and that it binds to the inward-open conformation is convincing.

Response:

We thank the reviewer for the support of our work and the input that has improved the manuscript.

The outward-open state provides a basis to understand substrate specificity, but this seems to fall short. The main issue remains that the transporter is designated as a histidine transporter, yet the authors cannot detect binding of any histidine- or histidine-containing peptides in PHT1. The authors indicate this could be a limitation in the thermal-shift assay. While this may be possible, the di-histidine peptides shows the strongest stabilisation at of all the di-peptides to PepT2. To me, this indicates that histidine-contains peptides is not a limitation of the assay itself. Indeed, if it was a limitation, then it would somewhat undermine all the peptides screening carried out here.

Response:

Different biochemical assays use various biophysical principles and conditions. All reported transport assays for histidine uptake by PHT1 were conducted *in vivo* and so far, direct binding of histidine in an *in vitro* setting has not been demonstrated. Apart from that, thermal stability assays, as many other assays, have certain limitations. In the case of membrane proteins, the proteins are typically extracted into detergent micelles, that can influence the internal dynamics and can have detrimental effects on binding substrates.

To identify potential substrates or inhibitors of proteins, often large libraries have been screened with different biophysical methods and the mutual overlap of individual hits across the different methods (including thermal shift assays) is often not so high. We have seen such a behavior also in the case of HsPepT2. In our binding assay, peptides with negatively charged residues (such as EE) are not binders. However, in a whole cell transport assay, peptides with negatively charged residues were reported to compete with a fluorescent reporter, including the EE dipeptide (see binding data presented in this manuscript and transport competition data in Killer M *et al.*, 2021 Science Advances). Therefore, we interpret a stabilization effect upon ligand binding in the thermal-shift assay as an indication for substrate interaction while we cannot exclude that a molecule that does not cause a thermal-shift does not interact.

The authors has tried to monitor uptake of histidine in PHT1 targeted to the plasma membrane, but due to background level being too high they cannot come to any firm conclusions. I am not sure what the other studies have done differently? As far as I am aware, I thought that PepT1/PepT2 were not capable of transporting free amino acids? Given the close structural similarity of PHT1, is there any indication that could enable PHT1 to transport free amino acids in addition to di-peptides?

Response:

Other studies describing histidine transport by PHT1 were typically performed in different cell types (*Xenopus* oocytes, COS-7 cells or MDCK cells), which could be the difference regarding the high background level that we have encountered.

Based on our own and literature data, PepT1 and PepT2 and various bacterial homologues are not capable of transporting single amino acids. The arrangement of the substrate binding pocket between PHT1 and PepT1 (or PepT2) is highly conserved, with the exception of Asp381 (corresponding to an asparagine residue in human PepT1 and human PepT2). In a recent publication from our lab, we studied the substrate selectivity of the bacterial peptide transporter DtpB by binding and transport assays and also determined 14 X-ray structures with different di- and tripeptides (Kotov et al., Cell reports 2023). From this study, we learned that the binding site of these transporters is not static, but in fact shows high plasticity and the binding site can adapt to accommodate di- and tripeptides with different sidechains. In analogy, the PHT1 binding site may have a similar or even greater plasticity than DtpB and is potentially able to accommodate a single Histidine and different di- and tripeptides.

The asparagine residue in PepT1/2 contributes to the stabilization of the N-terminus of di- and tripeptides and was shown to be critical for transport in other POT members. Asp381 in PHT1 is in the equivalent position and could further stabilize the N-terminus of peptides. In the case of a single amino acid, such as, histidine, it might compensate for a poorer interaction of the carboxylic group with the distant Arg44 and Lys180 residues located in the N-terminal domain. The histidine imidazole ring could interact with Asp381. However, structures of PHT1 with different substrates (Histidine, di- and tripeptides) are necessary to confirm such a hypothesis.

From my point of view, it therefore seems inconsistent to have a mechanism of PHT1 based on histidine transport when this cannot be confirmed here. I would prefer that the authors are clearer in the introduction and mechanism figure. It might be more prudent to outline that there data does not support the binding of histidine or histidine-containing peptides and as, such, how PHT1 is activated and linked to TASL needs further validation.

Response:

Histidine/Peptide-Transport is uncoupled from the signaling function of PHT1. To avoid any confusion, we have adjusted this in the figure and introduction. In the introduction and mechanism figure we now show the representation of the transport of peptides and protons by PHT1. We also included a small figure legend in the figure itself to make this clear.

We have also adjusted the discussion part to further strengthen the fact that we have not observed binding of histidine or histidine containing peptides in our binding assay. In line 384, you can now read: “Contrary to the plasma membrane peptide transporters, PHT1 (and PHT2) is known for its capacity in transporting histidine. However, the binding of histidine or histidine containing peptides was not observed in our assay. This could be attributed to an intrinsic limitation of the thermal shift assay, where although binding occurs, this is not translated in a thermal shift. This has been observed for fragment screening campaigns, using different biophysical methods, where some of the positive hits were not identify in the thermal shift assay^{52,53}. Therefore, other binding and transport assays are required prior any further conclusions can be drawn.”

In the discussion section, we now also mention a recent paper from Zhang et al. (Cell Rep. 42, 2023) that reports a study where their data are in agreement with our model and we have also included a short statement about future work regarding the PHT1-TASL pathway. In line 440, you can now read: “During the revision of this paper, a study by Zhang et al.⁵⁷ was published highlighting that the PHT1 transport activity is not necessary for IRF5 activation and cytokine production, when TASL is tethered to the endolysosome. This study further validates our model and agrees with our data. Future work is necessary to understand what triggers TASL engagement with PHT1 and how this activation is linked to the TLR pathway upstream.”

Reviewer #2 (Remarks to the Author):

The authors responded to my comments. I have no further comments.

Response: We thank the reviewer for the support of our work and the input that has improved our manuscript.